# Spatio-Temporal Variations in Pollen Limitation and Floral Traits of an Alpine Lousewort (*Pedicularis rhinanthoides*) in Relation to Pollinator Availability

**DOI:** 10.3390/plants12010078

**Published:** 2022-12-23

**Authors:** Wenkui Dai, Anne Christine Ochola, Yongquan Li

**Affiliations:** 1College of Horticulture and Landscape Architecture, Zhongkai University of Agriculture and Engineering, Guangzhou 510225, China; 2Center of Conservation Biology, Core Botanical Gardens, Chinese Academy of Sciences, Wuhan 430074, China

**Keywords:** bumble bee, flowering stage, floral traits, lousewort, pollen limitation, spatial–temporal variation

## Abstract

Populations of the same plant species living in different locations but flowering at different times may vary in pollinator availability and floral traits. However, the spatial and temporal links between floral traits and pollination are rarely included in single studies. In this study, three populations of an alpine lousewort, *Pedicularis rhinanthoides* Schrenk subsp. *tibetica* (Bonati) Tsoong, were surveyed to detect the variations in floral traits and pollinator activity. We hypothesized that floral divergence was spatio-temporally correlated with pollen limitation (PL) in relation to pollinators. Sampled plants from each population were divided into three groups, according to flowering stage: early, peak, and late. Pollen-supplementation experiments and investigations into pollinators, reproductive success, and floral traits were conducted on the plants from the different flowering stages and across the populations. Our results showed that the extent of PL varies across populations and among flowering stages. Populations in which more pollinators were recorded displayed a lower extent of PL. Furthermore, the temporal differences in PL showed a similar pattern for the three populations; the plants from the peak flowering stage suffered slighter PL than those from the other two stages. Nevertheless, some of the floral traits displayed similar spatial and temporal patterns to the PL, while the others only varied among the populations spatially. The results indicated that the performance of floral traits in a particular spatial–temporal situation shows they are well adapted to the corresponding pollination environment, which might help plants to optimize their reproductive fitness under different abiotic factors.

## 1. Introduction

Flowering and fruiting are key events in the life history of plants, and both are critical to their reproductive success [1,2]. The evolution of plants’ reproductive strategies was previously thought to be regulated by both abiotic (sunlight, soil, water condition, etc.) and biotic factors (especially pollinator availability) [3,4]. Pollinator decline is among the leading ecological challenges in this changing world [5,6,7,8] and, consequently, many flowering plants suffer critical pollen limitation (PL)—a decrease in potential sexual reproduction because of inadequate pollen receipt [9,10]. Additionally, PL may occur not only when pollinators become rare or absent, but also when the pollinator fauna changes. Such changes can result in inefficient pollen transfer, the deposition of heterospecific pollen, increases in autogamous and/or geitonogamous pollen flow, etc. [11,12,13]. Pollen limitation can be detected by hand cross-pollen supplementation when plants produce more seeds than in open pollination [14]. Although PL has been shown to influence plants’ flowering strategies [15,16,17,18,19], this effect still needs illustration by more empirical evidence to enhance the knowledge on the interactions of plants with ever-changing environmental factors.

Pollinators’ visitation of flowers has been found to vary in space and time [20,21,22]. In particular, plants flowering in alpine habitats may suffer from extreme variations in pollination because of changing weather conditions over time and heterogeneity in the population-distribution location. Many alpine environmental factors may result in unfavorable pollination conditions, such as low temperatures, strong winds, and short growing seasons [23]. Therefore, pollinator availability was determined as the main factor limiting the reproductive output of alpine plants, providing the appropriate research resources for examining the causes and effects of PL [3]. For alpine plants, populations from different habitats may differ in their pollinator abundance and/or composition, which is attributable to variations in temperature, availability of floral resources, etc. [3,24]. In addition, the density of flowers within a patch may influence the number of flower visits [18,25,26,27]. In addition to the separation of location, populations at different flowering stages may also vary in their pollinator abundance and/or composition because of the changes in flower density. For example, populations at the peak flowering stage always have more plants flowering simultaneously compared to those at the early and late flowering stages. Nevertheless, only a few studies have aimed to demonstrate the pattern of PL by incorporating both the inter-population (plants located at different habitats) and intra-population (plants flowering at different times) effects on the pollination and reproductive success of alpine plants [28,29].

Spigler and Kalisz (2013) stated that floral traits might alter to cope with the changes in the pollination environment [30]. Within populations, phenotypic variation in floral traits is common, e.g., flower and display size [18,27] or spatial separation between anthers and stigmas [31,32]. These changes in floral traits benefit shifts in mating systems in terms of selfing vs. outcrossing through a long-term adaption to the pollination environment [33]. For instance, the prolonged floral longevity of plants in high altitudes may compensate for low pollinator activity by presenting the flower for a longer time [34]; meanwhile, a larger floral display may be beneficial in attracting pollinators [35]. Furthermore, variations in pollinators between populations were found to be correlated with differences in floral traits [36]. Nonetheless, whether and how the floral traits spatio-temporally vary with the alternation of the pollination conditions still needs further investigation for a better understanding of the plant-pollinator interactions [3,25,37,38]. 

*Pedicularis rhinanthoides* Schrenk subsp. *tibetica* (Bonati) Tsoong is a long-tubed *Pedicularis* species endemic to southwestern China [39,40,41]. The species produces red and nectarless flowers with corolla beaks and relatively long corolla tubes of about 16–25 mm. The plant forms 1–15 inflorescences, 10–60 mm tall. Each inflorescence of the plant has 2–20 flowers. The stigma is slightly exerted into the corolla beak, while the style is based on the ovary and passes through the long tube. The four anthers are contained in the twisted corolla beak, and the pollen grains are released from a cleft after a pollinator that visits the flower [42]. No auto-fertilization (AF) occurs for this self-compatible plant because of the strictly spatial separation between the anthers to the stigma. By contrast, selfing, including geitonogamy, is aided by the exclusive pollinators—bumble bees [42,43,44]. In this study, the lousewort was used to detect the patterns of PL and floral divergence across the plants from different habitats (inter-population) and the plants within a population but flowering at different stages (intra-population). We investigated the extent of the PL- and fitness-related floral traits of early-, peak-, and late-flowering plants in three different populations (abbreviated as NA, NC, and NH; see Table 1). We hypothesized that the floral divergence might be spatio-temporally correlated with the extent of PL in relation to pollinator availability under the influence of different abiotic factors.

## 2. Results

### 2.1. Pollination and Reproductive Success

The flowering density per 2 m × 2 m plot varied among different flowering stages (*F*_2,36_ = 12.452, *p* < 0.001), but there were no differences among populations (*F*_2,36_ = 1.036, *p* = 0.365). The main pollinators were two bumble-bee species, *Bombus festivus* (Figure 1A) and *B. friseanus* (Figure 1B). The two bumble bees and their visiting frequencies varied among the populations (*F*_2,170_ = 47.020, *p* < 0.001) and flowering stages (*F*_2,170_ = 4.424, *p* = 0.013) (see Table 1 and Table 2). Compared to populations NC and NH, population NA received the highest frequency of bumble-bee visitation. Moreover, for all the populations, the visiting frequency by bumble bees for the plants at the peak flowering stage was higher than those at the early and late flowering stages (Table 1). Consequently, the stigmatic pollen load, fruit set, and seed production per capsule varied among the populations (*F*_2,256_ = 23.477, *p* < 0.001; *F*_2,239_ = 12.413, *p* < 0.001; *F*_2,261_ = 4.122, *p* = 0.017, respectively) and flowering stages (*F*_2,256_ = 29.572, *p* < 0.001; *F*_2,239_ = 16.675, *p* < 0.001; *F*_2,261_ = 4.823, *p* = 0.008, respectively; see also Table 3). The plants at the peak flowering stage had the maximum amount of stigmatic pollen load (Figure 2A), the highest fruit set (Figure 2B), and seed production per capsule (Figure 2C). Meanwhile, compared to the plant individuals at the peak and late flowering stages, the early plants had insufficient stigmatic pollen load (Figure 2A), decreased fruit set (Figure 2B) and seed production (Figure 2C). The temporal fluctuations were evident in populations NC and NH with unreliable pollinator activity through the three flowering stages. In population NA, the fluctuation was neutralized to a certain extent because of the abundant pollinator activities (Figure 2A).

### 2.2. Pollen Limitation

The seed production per capsule varied between the open pollination and supplemental pollination treatments (*F*_1,475_ = 47.858, *p* < 0.001), among the populations (*F*_2,475_ = 4.648, *p* = 0.01), and among the flowering stages (*F*_2,475_ = 5.628, *p* = 0.004) (see Appendix A). The seed production after the supplemental pollination treatment increased in the plants from populations NC and NH, but not in those from population NA (Figure 3). The plants at the early flowering stages for all three populations generally suffered more severe PL than those at the peak/late flowering stages (Figure 3), especially in the populations NC and NH.

### 2.3. Floral Divergence

For all the tested floral traits, the ovules per flower and corolla-tube length were almost consistent for the plants from the different flowering stages across the three populations, whereas the other six floral traits demonstrated spatial or spatio-temporal variations (see Table 4). The flower longevity and floral display size were substantially different among the plants from the three flowering stages and the different populations (Figure 4A,B). Meanwhile, the flower-display size (number of synchronously blooming flowers) and the height of the inflorescence varied among the different populations rather than the different flowering stages (Figure 4C,D). Nonetheless, the stigma–anther distance and pollen grains per flower showed a great difference for the plants from the different populations and at different flowering stages (Figure 4E,F). Of the plants at different flowering stages in each population, those at the peak flowering stages had the shortest flower longevity (Figure 4A) and smallest display size (Figure 4B), and produced flowers with the shortest anther–stigma distance (Figure 4E), but with the highest pollen production (Figure 4F).

## 3. Discussion

### 3.1. Spatio-Temporal Pollen Limitation

Previous studies revealed that the sexual reproduction of alpine plants is overwhelmingly pollinator-dependent [3,4,19,34]. Our results for the lousewort also indicated that fruit set and seed production were highly linked to pollinators (mainly *Bombus festivus* and *B. friseanus*) and their visitation, which may have been caused by the abundance of the floral resources. Patches with high flower density may attract more pollinators [27,45,46], increasing visiting frequency, which may enhance reproductive success. For the three studied populations, population NA, located near a wide wet meadow, demonstrated the highest flower density and three more bumble-bee-pollinated plants that co-flowered adjacently (*Pedicularis siphonantha*, *P. longiflora*, *Astragalus flavovirens*, etc.); thus, it had the highest frequency of visitation by bumble bees. Consequently, the plants from population NA and the peak flowering stage of each of the populations displayed greater reproductive success (fruit set and seed production) than those from the other two populations and the early and late flowering stages.

Furthermore, the results of the pollen-supplementation experiments indicated that the PL in the lousewort demonstrated spatio-temporal variations. The plants at the peak flowering stages of all three populations had the greatest likelihood of reducing the outcross PL compared to the plants from the other two flowering stages. Meanwhile, the outcross PL for the three studied populations was lower in population NA than in populations NC and NH. The spatio-temporal variations in the PL of the lousewort were reflected by the abundance of pollinator resources for the plants in the different flowering stages and those from different populations [47]. Additionally, this spatio-temporal variation could potentially contribute to differences in floral traits, which may help plants to enhance their reproductive output under pollen limitation [12,15,16,48]. Furthermore, the variations in the floral traits and the extent of the PL demonstrated similar spatio-temporal patterns in this lousewort species, aiding in the understanding of the floral divergence and population differentiation at the intra- and inter-population levels.

### 3.2. Spatio-Temporal Variations in Floral Traits in Relation to Pollinators

Spatio-temporal differences in pollinator availability may affect plant–pollinator interactions [48] and adaptively incur variations in floral traits to maintain reproductive fitness under pollen limitation [35,49]. This may help both plants and pollinators to adapt to changes in environment [50], such as changes in the successional flowering stage within a growing season and the separation of geographic location. Our results revealed that the spatio-temporal floral divergence was highly correlated with the extent of the PL, and that adaptive variations in floral traits are beneficial in enhancing plant reproductive success under unstable pollination conditions with pollinator loss or decline.

The number of ovules is a more conservative floral trait than pollen production in the evolution of flowering plants’ breeding systems [51]. This was proven in a series of taxa [10,52,53]. In this study, the number of pollen grains per flower differed in space and time, while the ovules per flower remained unchanged. Moreover, there was a similarity in the temporal fluctuation in pollen production across the three populations; the plants at the peak flowering stage always had the highest pollen production to meet the peak pollination activities within the flowering season. A greater allocation of resources to reproductive flowers during a population’s peak blooming period may help to enhance reproductive fitness, since the flowers may make a higher contribution to reproductive success compared to those from early or late flowering periods [54]. 

Although a short corolla tube (short pistil length) can help to form a reliable reproductive mechanism by enhancing the possibility of pollen self-fertilization [55], there was almost no difference between the populations at different flowering stages and sites in this lousewort. This finding implies that the corolla-tube length in this lousewort species could be a long-term adaptation to maintain reproductive success in changing environments. This differs from another long-tubed lousewort (*P. siphonantha*, with a 30–70 mm corolla tube), which varies widely in corolla tube length, even within individual plants [55]. The two louseworts have very different inflorescence structures. *Pedicularis rhinanthoides* Schrenk subsp. *tibetica* (Bonati) Tsoong has a very short raceme in which all flowers open on the same plane; an appropriate corolla-tube length is essential for a flower to display on the plane. For *P. siphonantha*, the long raceme allows the flowers with varied corolla-tube lengths to open at different positions on the inflorescence [52]. 

The results revealed that the floral traits of stigma–anther distance, floral longevity, and display size per inflorescence differed in the plants from the three flowering stages within a population and across the three populations. Therefore, it is proposed that this lousewort species’ three floral traits were highly phenotypically plastic [27,35,36]. The stigma–anther distance is thought to be linked with the pollen release and body size of the visiting bumble bee [52]. The variations could be adapted to the differences in the pollination environment among the different flowering stages across the populations. Floral longevity governs important reproductive processes influencing pollination and mating and varies considerably among angiosperm species; a flower opens within an allocated period until it receives enough conspecific pollen grains and then wilts rapidly [56]. In this lousewort species, individuals from the peak flowering stage and in a population with greater pollinator availability had shorter floral longevity [27,34,35]. An increase in inflorescence display size poses a dilemma because it should be helpful in pollinator attraction but might bear the genetic cost because of geitonogamous mating [3,34]. In this study, the inflorescence display size was larger in the stages or populations with lower pollinator availability but smaller in those with greater bumble-bee activity, which clearly reflected the selection under the display-size dilemma. However, the other two floral traits, namely the inflorescence height and the total number of flowers per inflorescence, varied among the populations rather than the flowering stages. Thus, the two floral traits were not influenced by the pollination environment but depended on genetics and/or the availability of abiotic resources [57]. The comparison of inter- and intra-population variations in floral traits herein should help to understand the pollination adaptation of plants flowering in different environments; the performance of floral traits in a particular spatial–temporal situation could help to enhance plant reproductive fitness through plant–pollinator interactions under the influence of different abiotic factors [3,4,16,20,22,27,33,58].

## 4. Materials and Methods

### 4.1. Study Species and Sites

*Pedicularis rhinanthoides* is widely distributed across the Hengduan Mountains and Tibet Plateau, which is the center for the diversification of the genus *Pedicularis* L. (Orobanchaceae) [39,40,41]. The alpine plant inhabits diverse habitat types (e.g., meadows, shrubs, and forest edges), and is distributed at altitudes from 3000 to 4500 m above sea level; it flowers from early July to late August and bears fruits from late August to September. We conducted our investigations on three populations (NA, NC, and NH, see Table 1) at different habitats in Shangri-La Court of Northwest Yunnan, south-western China, in 2013–2015.

We conducted the pilot study in 2013 to determine the experiment sites. We selected three populations distant from each other in wet shrubs or meadow edges. Each studied population had at least 300 plant individuals in areas measuring about 50 m × 50 m with some similar co-flowering plants, especially in population NA, where the other two *Pedicularis* species (*P. siphonantha*, *P. longiflora*) and *Astragalus flavovirens* blossomed in large numbers and shared in the local bumble-bees pollination. We divided the flowering of a population into three stages, according to Dafni [59]. The early flowering stage is when 25% of the individuals are flowering, the peak flowering stage is when at least 50% of the individuals are flowering, and the late flowering stage is when less than 50% of the individuals are flowering [59]. Subsequently, we conducted field investigations at three flowering stages for each population, mainly in 2014 and 2015. 

### 4.2. Pollination Observation

To compare pollinators’ activity among plants at different flowering stages from the three populations, we conducted pollination observations from 0900 h to 1700 h on 2–3 sunny days. We randomly selected fifteen plots, each with an area of 2 m × 2 m, from each population at each sampled flowering stage. We recorded the number of flowering inflorescences from these plots and averaged them to a single plot, which we used to represent the flowering density. On the established plots, we recorded the pollinator composition and their visiting frequency every 30 min and total over 15 h (one hour for each plot) for one population at each early/peak/late flowering stage. We captured the bumble-bees visiting the flowers for identification and released them to the field when it was a known species. Otherwise, we collected them for identification at the Institute of Zoology, Chinese Academy of Sciences. To evaluate the differences in pollinator composition at different flowering stages across the three populations, we recorded each bumble-bee species within the observation periods. We estimated the visiting frequency by the number of bumble bees visiting a plot within the 30-minute observational period.

In addition, we also counted the stigmatic pollen load to explore the pollination efficiency of each of the flowering stages from all three populations. For each population at the sampled flowering stage, we randomly selected no less than 30 flowers, each from different individuals, from the entire population and harvested them when the corolla wilted. We picked up the selected flowers and fixed them into FAA solution (formalin 37–40%, acetic acid, and ethyl alcohol at a ratio of 5:6:89 by volume) for counting pollen grains on the stigma in the laboratory. We treated the stigmas with 8 mol/L NaOH for 20 h and then observed them under a microscope (200× Nikon 80i, Nikon, Tokyo, Japan). 

### 4.3. Estimation of Reproductive Success and Pollen Limitation

The levels of reproductive success of flowers from different flowering stages across the three populations were compared. We randomly marked about 30 flowers with 20–25 inflorescences, each from different plant individuals at a particular flowering stage of the studied populations, from the whole population, and harvested them for the counting of the fruit set and seed production when the fruits were fully mature. Fruit set (%) was defined as the capsules with seeds divided by the total number of flowers for each inflorescence (20–25 for each population at certain flowering time). We counted the seed production per capsule and compared it between the different flowering stages across the studied populations after removing the empty ones from the 30 marked flowers. 

We conducted pollen-supplementation experiments (PSE) to detect the extent of PL for the plants at the different flowering stages from different populations. For every population at each flowering stage, we randomly selected 30 inflorescences of different individuals, and two flowers from each individual were marked before opening. One of the chosen flowers (30) was used for supplemental hand pollination with outcross-pollen collected from the plants 10 m away from each other, while the remaining one was under open pollination and treated as the natural control (30 flowers). The flowers were marked until the fruits were mature for the counting of seed production. Next, we calculated the PL according to the formula of Cosacov et al. [14]: PL = *F*_IX_/*F*_IN_ − 1(1)
where *F*_IX_ refers to the seed production by supplemental hand pollination with outcross-pollen and *F*_IN_ refers to that by open pollination.

### 4.4. Measurement of Floral Traits

To explore the flower divergence of the plants at different flowering stages from the three studied populations, we investigated floral traits, including floral longevity, floral display, size and height of the inflorescence, corolla-tube length, stigma–anther distance, numbers of ovules, pollen grains per flower. At each flowering stage for all the populations, we randomly selected more than 30 flowers/inflorescences, each from different individual plants, from the whole population to measure the floral traits. Floral longevity meant the period from flower opening to corolla wilt. Floral display size was measured as the average daily number of synchronously blooming flowers per inflorescence during the pollination observation time. These inflorescences were used to measure the inflorescence size by recording the total number of flowers within an inflorescence and the height by noting the distance from the ground to the top flower of the inflorescence. Corolla-tube length referred to the distance between the base of the corolla beak and the floral receptacle, and stigma–anther distance was measured from the tip of the stigma to the middle point of the area on the corolla beak where the anthers were located. Furthermore, we picked the other 30 flowers from different individual plants without dehiscing anthers. We then fixed them into formalin–aceto–alcohol (FAA) solution (3.7% formaldehyde, 5% acetic acid, 50% ethanol) for counting the number of pollen grains and ovules per flower later, in the laboratory, under a microscope (Nikon 80i, 200×, 40×, respectively). 

### 4.5. Data Analysis

To detect the spatio-temporal differences in the pollinators’ activities, covariance analysis was used to compare the visiting frequencies among plants from the different flowering stages (fixed factor I) and the three studied populations (fixed factor II), with main effect and interaction, and the flowering density as the covariant. When the difference was significant, we conducted a multi-comparison among the different flowering stages and populations. 

A two-way ANOVA was used to detect the spatio-temporal variations in the stigmatic pollen load, fruit set, seed production and/or each test floral trait among the plants from the different populations (fixed factor I) and flowering stages (fixed factor II), with main effect and interaction. Multi-comparison of those parameters was conducted among the different flowering stages and populations. 

We conducted multi-ANOVA to detect the effects of pollination treatment, flowering stage, and population on the seed production per capsule (fixed model with main effect and interaction) for PSE. Additionally, the independent T-test was used to detect whether adding extra pollen increased seed production per capsule.

Data processing and variance analyses were conducted by Excel 2010 (Microsoft, Redmond, WA, USA) and SPSS 21.0 (SPSS Inc., Chicago, IL, USA). We tested all the data by normal distribution or conducted data conversion before analysis of variance. The significance level was defined at *p* = 0.05. 

## 5. Conclusions

This study incorporated spatial and temporal variations in PL and floral traits by investigating plants from three populations and plants flowering at different stages within each of these populations. The reproductive success was highly dependent on pollinator activity, demonstrating the varied extent of the PL. Moreover, the floral divergence was positively correlated with PL. Some of the floral traits varied spatially and temporally, while the others only displayed variations in spatial scale. However, both variations proved to be well adapted to pollinator availability. The finding that the spatial–temporal variations in floral traits and PL are linked to pollinator availability help to explain the alternation in plant reproductive strategies the face of pollinator decline in changing environments.

## Figures and Tables

**Figure 1 plants-12-00078-f001:**
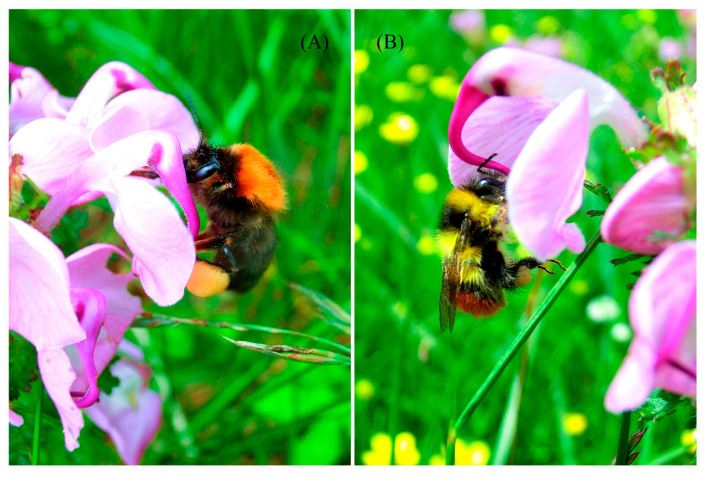
Main bumble-bee species visiting the flowers of *Pedicularis rhinanthoides* Schrenk subsp. *tibetica* (Bonati) Tsoong: (**A**), *Bombus festivus*; (**B**), *B. friseanus*.

**Figure 2 plants-12-00078-f002:**
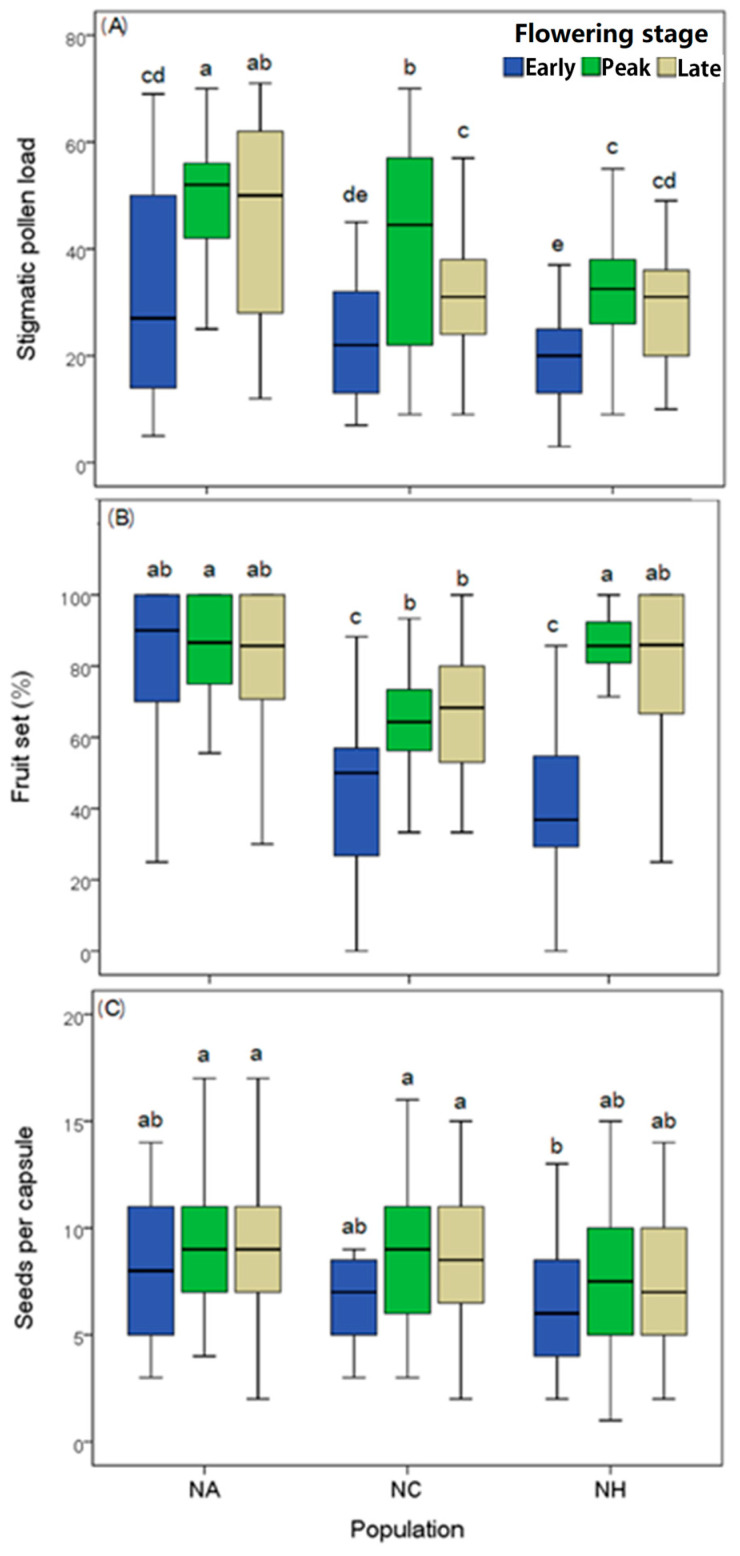
Comparison of stigmatic pollen load (**A**), fruit set (**B**), and seeds per capsule (**C**) for plants from different flowering stages across the three populations. Lower-case letters indicate the results of multiple comparisons.

**Figure 3 plants-12-00078-f003:**
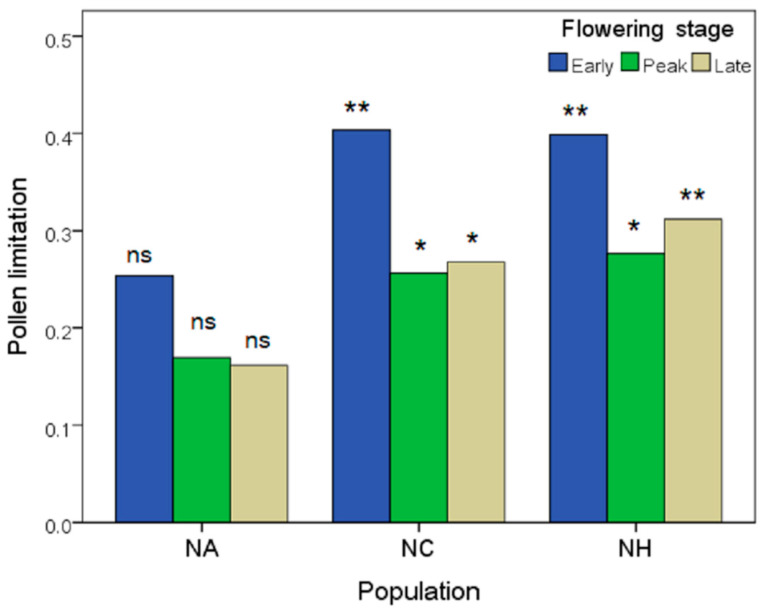
Comparison of pollen limitation (PL index) for plants from different flowering stages across the three populations. Bars with stars refer to the differences between the pollination treatments for each population at a certain flowering stage. Significant level is at *p* = 0.05: no significance (ns), *p* ≥ 0.05; * *p* < 0.05; ** *p* < 0.01.

**Figure 4 plants-12-00078-f004:**
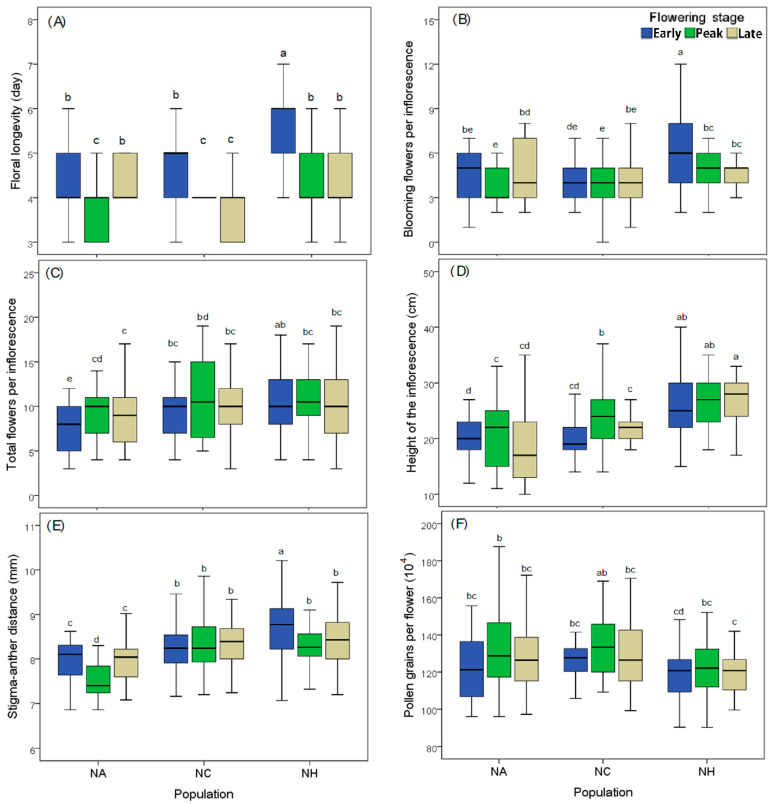
Comparison of floral traits selected in this study for plants at different flowering stages across the three populations, namely, flower longevity (**A**), inflorescence display size (**B**), the total number of flowers per inflorescence (**C**), inflorescence height (**D**), stigma–anther distance (**E**), and pollen production per flower (**F**). Lower-case letters indicate the results of multiple comparisons.

**Table 1 plants-12-00078-t001:** The information on the three studied populations at three flowering stages.

Study Population	Location	Flowering Stage	Flower Density (Mean ± SE, 2 m × 2 m)	Main Pollinators (*Bombus*)		Visiting Frequency (Mean ± SE, 2 m × 2 m, 30 min)
				*B. friseanus*	*B. festivus*	
NA	99.71° E, 27.96° N; 3390 m	Early	10.2 ± 1.7	+	-	0.195 ± 0.029
		Peak	24.8 ± 3.6	+	+	0.285 ± 0.048
		Late	12.0 ± 1.3	+	+	0.175 ± 0.035
NC	99.80° E, 27.82° N; 3300 m	Early	11.4 ± 3.6	-	-	-
		Peak	19.4 ± 2.4	+	+	0.145 ± 0.035
		Late	9.2 ± 1.1	+	+	0.065 ± 0.023
NH	99.75° E, 27.62° N; 3280 m	Early	11.0 ± 1.7	-	-	-
		Peak	22.3 ± 5.8	+	-	0.035 ± 0.011
		Late	16.8 ± 2.6	+	-	0.015 ± 0.008

NA, NC, and NH are abbreviations of the study sites; “+” refers that the bumble bee species has been recorded, while “-“represents none.

**Table 2 plants-12-00078-t002:** ANCOVA of the effects of population and flowering stage (fixed factors), with flower density as the covariant, on bumble-bee visiting frequency.

Variable	Bumble-Bee Visiting Frequency (2 × 2 m^2^, 30 min)			
Effect	*df*	*MS*	*F*	*p*
Density (covariant)	1	0.549	0.393	0.531
Population	2	65.635	47.020	<0.001
Stage	2	6.175	4.424	0.013
Population × stage	4	2.347	1.681	0.157
Error	170	1.396		
Total	180			

**Table 3 plants-12-00078-t003:** Two-way ANOVA of the effects of population (fixed factor I) and flowering stages (fixed factor II) on stigmatic pollen load, fruit set, and seeds per capsule.

Variable	Stigmatic Pollen Load				Fruit Set (%)				Seeds per Capsule			
Effect	*df*	*MS*	*F*	*p*	*df*	*MS*	*F*	*p*	*df*	*MS*	*F*	*p*
Population	2	4941.358	23.477	<0.001	2	8359.227	12.413	<0.001	2	44.446	4.122	0.017
Flowering stage	2	6224.111	29.572	<0.001	2	11,229.09	16.675	<0.001	2	52.009	4.823	0.009
Population × stage	4	112.286	0.533	0.711	4	1942.166	2.884	0.023	4	0.925	0.086	0.987
Error	256	210.476			239	673.406			261	10.784		
Total	265				248				270			

**Table 4 plants-12-00078-t004:** Two-way ANOVA of the effects of population (fixed factor I) and flowering stages (fixed factor II) on floral traits selected in this study.

**Variable**	**Floral Longevity (Day)**	**Floral Display**	**Size of Inflorescence**
**Effect**	** *df* **	** *MS* **	** *F* **	** *p* **	** *df* **	** *MS* **	** *F* **	** *p* **	** *df* **	** *MS* **	** *F* **	** *p* **
Population	2	10.293	20.686	<0.001	2	60.044	22.505	<0.001	2	157.315	12.274	<0.001
Flowering stage	2	18.426	37.032	<0.001	2	18.433	6.909	0.001	2	19.937	1.556	0.213
Population × stage	4	4.004	8.046	<0.001	4	12.844	4.814	0.001	4	32.587	2.543	0.04
Error	261	0.498			261	2.668			261	12.817		
Total	270				270				270			
**Variable**	**Height of Inflorescence (cm)**	**Stigma–Anther Distance (mm)**	**Pollen Grains per Flower (10^4^)**
**Effect**	** *df* **	** *MS* **	** *F* **	** *p* **	** *df* **	** *MS* **	** *F* **	** *p* **	** *df* **	** *MS* **	** *F* **	** *p* **
Population	2	1626.844	71.186	<0.001	2	17.25	68.456	<0.001	2	24.744	8.357	<0.001
Flowering stage	2	62.878	2.751	0.066	2	4.091	16.233	<0.001	2	9.12.0	3.08	0.048
Population × stage	4	103.322	4.521	0.002	4	1.638	6.502	<0.001	4	0.797	0.269	0.898
Error	261	22.853			261	0.252			261	2.961		
Total	270				270				270

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
