# Peer review of "Spatio-Temporal Variations in Pollen Limitation and Floral Traits of an Alpine Lousewort (Pedicularis rhinanthoides) in Relation to Pollinator Availability"

_plants, 2022, doi:10.3390/plants12010078_

Round 1

Reviewer 1 Report

My comments are in the attached Word file.

Reviewer 2 Report

In this study,  three populations of an alpine lousewort, Pedicularis rhinanthoides Schrenk subsp. tibetica (Bonati) Tsoong, were used to detect interaction of spatio-temporal variations in floral traits and pollinators’ availability. It is a very interesting topic especially due the expected effects of climate change. However, the scope of the study should be clearly stated in the abstract and introduction (hypothesis made) and reflected in the title. Also the discussion should be aligned with the hypothesis. Overall good work to be used as a model for other interactions of plant-pollinator interactions. Overall, very good work to be used as a model for other interactions of plant-pollinator interactions.Please see special comments in the uploaded file.

Round 2

Reviewer 1 Report

Thank you for the detailed text corrections and responses to my comments. I found three more things to improve:  

176-177 Please explain in the text why the results for NA population are different compared to the other two population.

Moreover, add in the Discussion section what preferences of the recorded bumblebee species are. Please check if you have provided all the explanations about the methods used.

345 Correct  "ulti-Anova" to multi-Anova
